# Presence and Distribution of Fluoride Ions in Groundwater for Human in a Semiconfined Volcanic Aquifer

**Cecilia Judith Valdez-Alegría [1], Rosa María Fuentes-Rivas [2], José Luis García-Rivas [1], Reyna María Guadalupe Fonseca-Montes de Oca [3] and Beatriz García-Gaitán [1,\*]**

1   Tecnológico Nacional de México/Instituto Tecnológico de Toluca, Av. Tecnológico s/n, Colonia Agrícola Bellavista, Metepec 52149, Estado de México, Mexico; cvaldeza@hotmail.com (C.J.V.-A.); joseluisgarciarivas279@gmail.com (J.L.G.-R.)
2   Facultad de Geografía, Universidad Autónoma del Estado de México, Cerro de Coatepec s/n, Ciudad Universitaria, Toluca 50110, Estado de México, Mexico; rmfuentesr@uaemex.mx
3   Instituto Interamericano de Tecnología y Ciencias del Agua (IITCA), Universidad Autónoma del Estado de México, Unidad San Cayetano, Km.14.5 Carretera Toluca-Atlacomulco, Toluca 50200, Estado de México, Mexico; mgfonsecam@uaemex.mx
\*   Correspondence: beatrizggmx@yahoo.com; Tel.: +52-722-2087224

**Abstract:** Dental and emaciated fluorosis is derived from the chronic intake of fluoride ions ($F^-$) by consumption of food, tooth products and drinking water from groundwater. Recent reports indicate that in some regions of Mexico, the incidence of fluorosis in temporary and permanent dentitions have increased in recent years. The purpose of the present investigation was to determine the presence and distribution of $F^-$ ions in semi-confined aquifers, located in the basins of Lerma-Chapala and Valley of México. Temperature (T), pH, electrical conductivity (EC) and alkalinity were determined in situ, in 27 groundwater wells. The hardness, chloride ions ($Cl^-$), free chlorine ($Cl_2$), total dissolved solids (TDS) and bicarbonates ($HCO_3^-$), were determined in the laboratory. The high content of bicarbonate ions ($HCO_3^-$) is associated with the alteration of silicates and the presence of chloride ions ($Cl^-$) due to the anthropogenic activity. The highest value of fluoride ions ($F^-$) (1.31 mg/L) was identified in a well of the Lerma-Chapala basin and in 25% of the samples (1.056 mg/L) of the Valley of México basin. The groundwater in the Lerma-Chapala basin is of the $Ca-HCO_3$ type and the water in the Valley of Mexico basin is $Na-HCO_3$ and $Ca-HCO_3$. According to the Gibbs diagram, the water of the Toluca Valley aquifer is recharged by local meteoric precipitation and consequent the water-rock interaction occurs. The correlation analysis shows the significant relationship between $F^-$ and CE (R = 0.5933), $F^-$ and alkalinity (R = 0.6924), $F^-$ and $HCO_3^-$ (R = 0.6924) and $F^-$ and TDS (R = 0.5933). The correlations confirm that the content of fluoride ions present in groundwater is associated with high concentrations of bicarbonate ions and the presence of sodium, due to a process of alteration of the silicates.

**Keywords:** fluoride; groundwater; dental fluorosis; health risk; natural pollution; anthropogenic contamination

## 1. Introduction

Groundwater is the largest source of drinking water; its accessibility can be threatened not only by the introduction of pollutants from human activities, but also by natural processes [1]. The natural contamination of groundwater due to geological environments can be an important factor that limits its usage for human consumption. The high or low concentration of certain ions in the water is a

major problem, as they make it unsuitable for consumption. Fluoride is one of the ions that cause health problems. The weathering of the fluorine-rich rocks, fluorapatite ($Ca_5(PO_4)_3F$, fluorite ($CaF_2$) and, cryolite ($Na_3AlF_6$) and the infiltration of rainwater through them, increases the concentration of fluoride in the groundwater causing natural contamination. Fluorite is found in sedimentary rocks and cryolite is found in igneous rocks. These minerals are almost insoluble in water; therefore, they can only be present in groundwater when conditions favor their dissolution or when industrial effluents with high concentrations of $F^-$ ions are discharged into water bodies [2].

Contamination by fluoride ions ($F^-$) dissolved in groundwater, intended for human consumption, is an epidemic problem around the world [3,4]. The World Health Organization (WHO) recommends a maximum limit of $F^-$ ions content to 1.5 mg/L. The consumption of fluoride ions in concentrations significantly higher than this limit (1.5 mg/L), can cause the following problems: dental and skeletal fluorosis, osteoporosis, arthritis, infertility, brain damage, Alzheimer's and thyroid. In addition, it also increases the risk or susceptibility to kidney diseases and cancer [2,5–12]. It may also cause a deficient development of the human brain; reducing, among other effects, the intellectual coefficient (IQ) on school-age children [13].

At least 26 countries around the world, namely Argentina, Canada, Chile, the United States, Hungary, Italy, Pakistan, China, India, Kenya, Mexico, South Africa, Tanzania and Uganda, suffer from endemic fluorosis due to consumption of groundwater with high content of fluoride ions, originated largely from geological phenomena. The result of the natural interaction of the water with some of the volcanic rocks widely dispersed in the soil, that compose some of the main aquifers used as water supply for the population [3,4,14,15].

In Mexico, the main source of water supply for human consumption is from groundwater; in the states of Aguascalientes, Chihuahua, Durango, Mexico, Guanajuato, Jalisco, San Luis Potosi, Sonora, Zacatecas and Morelos. In certain areas, it has been detected that concentrations of fluoride ions in the water exceed the permissible limit according to the norm NOM-127-SSA1-2001 (1.5 mg/L); for this reason, five states of the above mentioned are excluded from the National Fluoridation of Salt Program (Programa Nacional de Flouracion de la Sal-PNFS) [16]. Valenzuela et al. [15] reported that over five million Mexican inhabitants are chronically exposed to high concentrations of fluoride ions through water consumption.

Alanís (1987), in the study carried out in 858 towns of the State of Mexico, reported and classified concentrations of fluoride ions into: low, medium, optimal and high, for the quantification in groundwater [17,18]. Results were: 672 (78.32%) water samples showed low concentration (0.0 to 0.29 mg/L), 167 (19.46%) samples average concentration was observed (0.3 to 0.69 mg/L), 13 locations showed an optimum concentration of (0.7 to 0.99 mg/L) and, finally, high concentration were found in six sites (1.0 to 1.5 mg/L). The municipality of Cuautitlán Izcalli, in the town of San Francisco Tepojaco, had an optimum concentration of 0.75 mg/L. In the municipality of San Felipe del Progreso, low concentrations were found in both the well bearing the same name of the municipality and in the San Agustín well (0.26 mg/L and 0.17 mg/L, respectively). In the municipality of Tenango del Valle, in the localities of San Francisco Putla, San Pedro Tlanixco and Santiago Cuautzenco, low concentrations (0.18 mg/L, 0.09 mg/L and 0.22 mg/L, respectively) were found.

On the other hand, in the municipality of Toluca, in the town of San Buenaventura, San Lorenzo Tepaltitlán and San Mateo Oxtotitlán, low concentrations were reportedly 0.06 mg/L, 0.18 mg/L and 0.10 mg/L respectively; while in the municipality of Zumpango de Ocampo, in the town of San Juan Zitlaltepec, a high concentration of 1.0 mg/L of fluoride ions was reported in the water. Ten years later, in 1997, the Institute of Health of the State of Mexico (ISEM) conducted a study of groundwater intended for human consumption: the physicochemical analysis laboratory of the National Water Commission (CONAGUA) reported that in the municipalities of Cuautitlán Izcalli, Tenango del Valle and Zumpango de Ocampo, the water presented a high content of $F^-$, which means that the children of those municipalities are exposed to a chronic intake of $F^-$ during their dental growth, causing dental fluorosis.

The purpose of this study was to determine the presence and distribution of fluoride ions in the groundwater for human use and consumption in six municipalities of the State of Mexico, four located in the aquifers within the limits of the Lerma-Chapala Basin and two in the basin of the Valley of Mexico, and to establish the health risk from its consumption.

## 2. Materials and Methods

### 2.1. Study Area

The study area is located within the limits of the Lerma-Chapala basin and the Valley of Mexico basin. The region of the Lerma-Chapala basin is located in the center west of the Mexican republic with an extension is 53,591 km$^2$, which is equivalent to 3% of the country's territorial extension. It houses 11% of the population and includes territories of five states: State of Mexico (9.8%), Querétaro (2.8%), Guanajuato (43.8%), Michoacán (30.3%) and Jalisco (13.4%) [19]. The water from the Lerma-Chapala basin supports human as well as agricultural and industrial needs. Within this basin, specifically in the jurisdiction of the State of Mexico, are the aquifers Valley of Toluca, Ixtlahuaca-Atlacomulco, Tenancingo and Villa Victoria-Valley of Bravo; these aquifers feed the underground zone of the municipalities of El Oro, San Felipe del Progreso, Tenango del Valle and Toluca (Figure 1).

The Valley of Mexico Basin is located in the center-south of the Transverse Neovolcanic Axis, has an average altitude of 2240 m above sea level and an approximate area of 9600 km$^2$. It is limited to the north by the Sierra de Pachuca and the Sierra de Tezontlalpan, to the south by the Sierra de Chichinautzin, to the east by the Sierra Nevada, the Sierra Calpulalpan and the Sierra de Tepozán and to the west by the Sierra of Monte Bajo and the Sierra de las Cruces. The aquifer 1508 is found in this basin (Cuautitlán-Pachuca), which feeds the underground zone of the municipalities of Cuautitlán Izcalli and Zumpango de Ocampo (Figure 1).

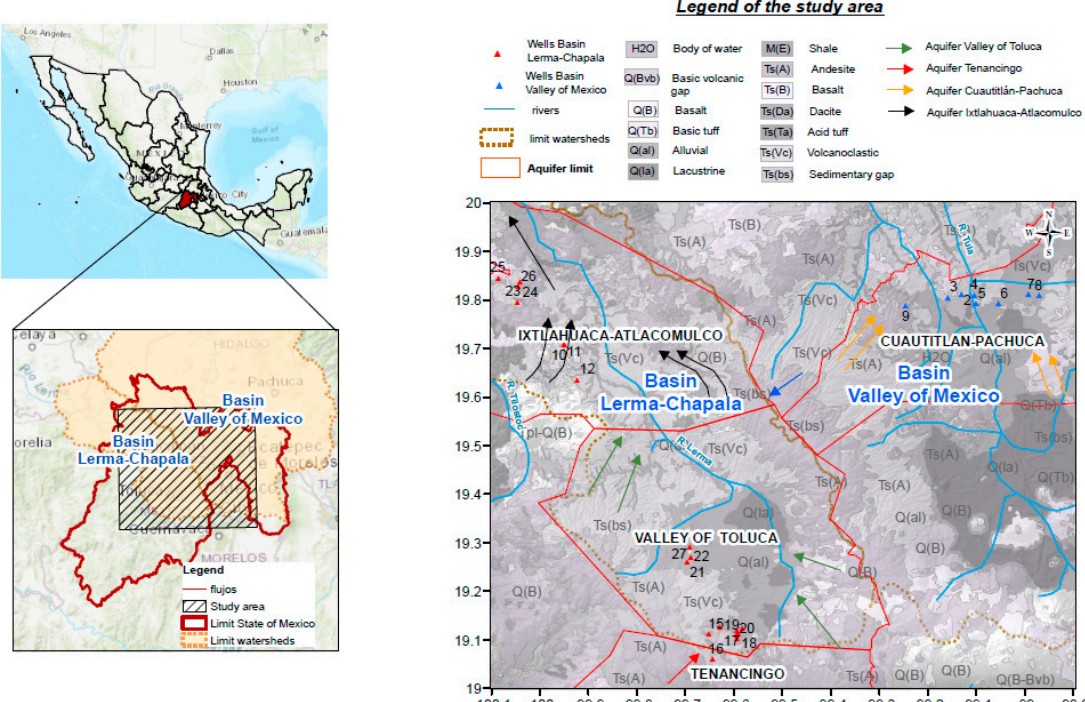

**Figure 1.** Location of the study area and location of the sampling points within the limits of the Lerma—Chapala and Valley of México basins.

According the porosity the Toluca Valley aquifer is divided in two types: porous and fractured. The porous aquifer is built up from unconsolidated clastic deposits, which fill the basin of the Toluca Valley, represented by tuffs, sands, alluvial deposits (Qal) and lacustrine deposits (Qla) of the Tarango

formation. The aquifer with fractured porosity is build up from andesites (Tpv) and ancient basalts. With these porosity geological properties, the aquifer is integrated as a free upper aquifer and the aquifer with fractured medium as a semiconfined lower aquifer. Even when both layers are divided by impermeable materials, they are considered part of the same aquifer [20,21]. As for the aquifer flow system of the Toluca Valley, it is formed by three main directions, the first originates from the Sierra de las Cruces towards the foothills west-circulation, the second comes from the Nevado de Toluca North direction with exit at Ixtlahuaca, and the third originates from the infiltrations of the Nevado east-northeast direction [20]. This aquifer is characterized by a non-totally impermeable top, formed by aquitards, which allow vertical water infiltration so it can receive recharges or lose water through the top or the base [22,23], the aquifer is located in the physiographic region known as the Trans-Mexican Neovolcanic Belt [22,24].

*2.2. Sampling*

Within the limits of the basins mentioned before, 27 sampling sites were located, mainly in the municipalities of Cuautitlán Izcalli, El Oro, San Felipe del Progreso, Tenango del Valle, Toluca and Zumpango de Ocampo (Figure 1). The sampling was carried out during the rainy season, the sampling for the quantification of: fluoride ions, temperature, pH, electrical conductivity (EC), alkalinity, hardness, chlorides ($Cl^-$), free chlorine ($Cl_2$) and bicarbonates ($HCO_3^-$), as well as its preservation, was carried out following the procedures established in the norm NOM-230-SSA1-2002 and [25]. For the collection of the samples polyethylene bottles were used previously submerged in an acid solution during an hour and rinsed with deionized water. After collection and transport, the samples were kept refrigerated at 4 °C until their analysis in the laboratory [25].

The physicochemical parameters in situ were determined for each of the samples: pH, temperature (T) and electrical conductivity (EC). The samples were measured in triplicate and consecutive way, an average value of the measurements was obtained [25]. The quantification of fluoride ions ($F^-$), of each of the samples, was performed three times using the potentiometric method, according to the procedure established in the standard NMX-AA-077-SCFI-200. Likewise, the physicochemical parameters, alkalinity, hardness, $Cl^-$, $HCO_3^-$ and $Cl_2$ were carried out following the procedures set by [25].

## 3. Results and Discussion

Table 2 shows the results of the physicochemical parameters obtained in situ: temperature, pH and electrical conductivity, of the 19 water samples collected in the Valley of Toluca aquifer and the Tenango del Valle aquifer, both located within the limits of the Lerma-Chapala basin; it also shows the concentrations of fluoride ions ($F^-$), chloride ions ($Cl^-$), alkalinity ($CaCO_3$), hardness ($CaCO_3$) and bicarbonates ($HCO_3^-$). The majority of physicochemical in situ parameters showed great spatial variation, except Temperature and pH. The low values of temperature and pH were identified near the recharge zones (Xinanteclatl Volcano, Nevado de Toluca) and increase in the direction of the groundwater flow. According to the determined pH values, the analyzed water is considered neutral to slightly alkaline, the pH values ranged between 6.99 and 7.96, with an average value of 7.51; the values of pH and temperature found in the Valley of México basin (Table 1) ranged between 7.0 and 7.8 and from 21.8 °C to 30.2 °C, respectively. The temperature of the water in the Valle of Mexico basin is higher, by 5 °C, than the temperature reported for the samples from the Lerma-Chapala basin, where the maximum value was 25.5 °C, the minimum of 13.8 °C and the average of 19.15 °C.

In relation to the values of electrical conductivity, significant spatial differences were observed (Figure 2). The highest EC value was found in well 11 (609 µS/cm) located in the municipality of San Felipe del Progreso, a shallow well (72 m), which indicates that the water is more mineralized at this point and that it may be receiving infiltrated surface water. The EC values ranged between 99 and 609 µS/cm, while the mean value was 268.47 µS/cm. The highest values were observed in the municipalities of San Felipe del Progreso, Tenango del Valle, El Oro and Toluca, the values increase

towards the central part of the Lerma-Chapala basin, parallel to the groundwater flow (Figure 2). Meanwhile, the highest value of EC, observed in the Valley of México basin in the municipality of Zumpango de Ocampo, corresponds to well 5 (1642 µS/cm) and the lowest value in the municipality of Cuautitlán Izcalli, in well 9 (492.3 µS/cm).

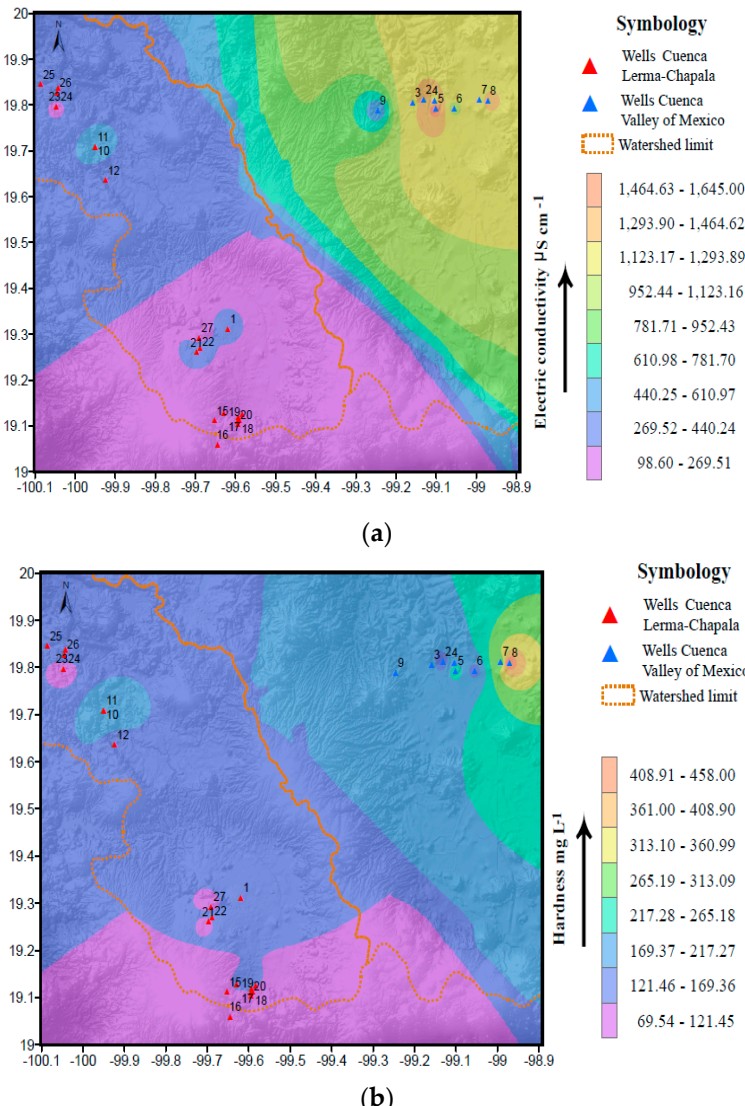

**Figure 2.** (**a**) Spatial distribution of electrical conductivity of 27 groundwater samples from the Lerma—Chapala and Valley of México basins; (**b**) Spatial distribution of hardness of 27 groundwater samples from the Lerma—Chapala and Valley of México basins.

**Table 1.** In situ, physicochemical and fluoride ions parameters of water samples from the wells located in the Valley of Mexico basin.

| ID | Depth | T | pH | EC | F⁻ | Cl⁻ | Alkalinity | Cl₂ | Hardness | HCO₃⁻ | TDS |
|---|---|---|---|---|---|---|---|---|---|---|---|
|  | (m) | (°C) |  | (µS/cm) | (mg/L) | (mg/L) | (mg/L) | (mg/L) | (mg/L) | (mg/L) | (mg/L) |
| MPL |  |  | 6.5–8.5 |  | 1.5 | 250 | 300 | 0.2–1.5 | 500 | N.E | 1000 |
| 2 | 110 | 23.9 | 7.0 | 1388 | 1.06 | 79.9 | 538 | S/C | 129 | 656 | 888.3 |
| 3 | 82 | 25.1 | 7.6 | 1237 | 1.01 | 62.4 | 574 | 0.7 | 179 | 700 | 791.7 |
| 4 | 90 | 24.5 | 7.8 | 1230 | 0.79 | 79.9 | 452 | S/C | 204 | 551 | 786.2 |
| 5 | 120 | 27.2 | 7.5 | 1642 | 0.70 | 124.8 | 600 | S/C | 244 | 732 | 1050.9 |
| 6 | - | 30.2 | 7.8 | 1053 | 0.51 | 99.4 | 372 | S/C | 149 | 454 | 673.9 |
| 7 | - | 23.1 | 7.8 | 1193 | 0.39 | 60.4 | 413 | S/C | 269 | 504 | 763.5 |
| 8 | - | 26.2 | 7.3 | 1346 | 0.29 | 36.6 | 641 | S/C | 458 | 782 | 861.4 |
| 9 | - | 21.8 | 7.7 | 492 | 0.36 | 13.7 | 335 | S/C | 189 | 408 | 315.1 |
| max | 120 | 30.2 | 7.8 | 1642 | 1.06 | 124.8 | 641 | 0.7 | 458 | 782 | 1050.9 |
| min | 82 | 21.8 | 7.0 | 492 | 0.29 | 13.7 | 335 | 0.7 | 129 | 408 | 315.1 |
| average |  | 24.94 ± 2.31 | 7.57 ± 0.23 | 904.75 ± 509.8 | 0.51 ± 0.31 | 52.72 ± 37.65 | 370.17 ± 200.2 | 1.06 ± 0.56 | 192.67 ± 98.82 | 421.36 ± 231.78 | 766.4 ± 213.2 |

T.: Temperature; CE: Electrical conductivity; F⁻: Fluorides; Cl⁻: Chlorides; Cl₂: Residual chlorine; HCO₃⁻: Bicarbonates; STD: Total dissolved solids; MPL: maximum permissible limit, N.E: not specified.

**Table 2.** In situ parameters, physicochemical and fluoride ions of water samples from the wells located in the Lerma—Chapala basin.

| ID | Depth | T | pH | EC | F⁻ | Cl⁻ | Alkalinity | Cl₂ | Hardness | HCO₃⁻ | TDS |
|---|---|---|---|---|---|---|---|---|---|---|---|
| | (m) | (°C) | | (µS/cm) | (mg/L) | (mg/L) | (mg/L) | (mg/L) | (mg/L) | (mg/L) | (mg/L) |
| MPL | | | 6.5–8.5 | | 1.5 | 250 | 300 | 0.2–1.5 | 500 | N.E | 1000 |
| 1 | - | 19.9 | 7.4 | 284.0 | 0.17 | 25.4 | 181.0 | S/C | 159.0 | 221.0 | 181.6 |
| 10 | 96 | 25.5 | 7.7 | 391.0 | 0.26 | 0.0 | 290.0 | S/C | 159.0 | 354.0 | 249.9 |
| 11 | 72 | 21.2 | 7.4 | 609.0 | 0.49 | 7.8 | 387.0 | S/C | 249.0 | 472.0 | 390.0 |
| 12 | - | 23.1 | 7.9 | 281.0 | 0.17 | 3.9 | 146.0 | S/C | 149.0 | 178.0 | 179.9 |
| 13 | 137 | 19.1 | 7.2 | 217.0 | 0.23 | 7.8 | 130.0 | S/C | 119.0 | 159.0 | 138.9 |
| 14 | - | 18.0 | 7.0 | 258.0 | 0.12 | 0.1 | 130.0 | S/C | 133.0 | 159.0 | 165.3 |
| 15 | - | 18.0 | 7.2 | 157.0 | 0.18 | 3.9 | 125.0 | S/C | 87.0 | 152.0 | 100.5 |
| 16 | 300 | 18.0 | 7.3 | 99.0 | 0.13 | 7.8 | 103.0 | 1.0 | 103.0 | 126.0 | 63.14 |
| 17 | - | 20.0 | 8.0 | 307.0 | 0.16 | 11.7 | 146.0 | S/C | 189.0 | 178.0 | 196.7 |
| 18 | 100 | 16.2 | 7.3 | 167.0 | 0.29 | 11.7 | 144.0 | 0.3 | 77.0 | 176.0 | 106.6 |
| 19 | 300 | 16.5 | 7.8 | 162.0 | 0.23 | 7.8 | 139.0 | 0.3 | 129.0 | 169.0 | 103.7 |
| 20 | 200 | 13.8 | 7.9 | 264.0 | 1.31 | 7.8 | 114.0 | 0.5 | 70.0 | 139.0 | 169.1 |
| 21 | - | 18.8 | 7.5 | 287.0 | 0.21 | 16.9 | 118.0 | 0.8 | 101.0 | 143.0 | 183.9 |
| 22 | 250 | 19.3 | 7.5 | 300.0 | 0.06 | 25.7 | 102.0 | 1.0 | 172.0 | 124.0 | 192 |
| 23 | 115 | 25.4 | 7.4 | 206.0 | 0.06 | 10.3 | 87.0 | 1.3 | 87.0 | 106.0 | 132.0 |
| 24 | 140 | 24.4 | 7.6 | 307.0 | 0.32 | 12.5 | 132.0 | 1.0 | 117.0 | 161.0 | 196.6 |
| 25 | 200 | 25.6 | 7.6 | 340.0 | 0.36 | 24.8 | 132.0 | 1.0 | 127.0 | 161.0 | 217.6 |
| 26 | 200 | 21.9 | 7.7 | 423.0 | 0.21 | 27.9 | 166.0 | 1.3 | 160.0 | 202.0 | 270.9 |
| 27 | 200 | 19.9 | 7.5 | 244.0 | 0.019 | 2.3 | 156.0 | 0.0 | 98.0 | 191.0 | 155.9 |
| max | 300 | 25.6 | 8.0 | 609.0 | 1.31 | 27.9 | 387.0 | 1.3 | 249.0 | 472.0 | 390.0 |
| min | 72 | 13.8 | 7.0 | 99.0 | 0.02 | 0.0 | 87.0 | 0.0 | 70.0 | 106.0 | 63.1 |
| average | | 20.20 ± 3.36 | 7.51 ± 0.28 | 269.1 ± 113.67 | 0.30 ± 0.28 | 11.4 ± 8.83 | 154 ± 70.62 | 0.80 ± 0.51 | 130.8 ± 44.31 | 198.5 ± 89.89 | 178.7 ± 72.8 |

T.: Temperature; CE: Electrical conductivity; F⁻: Fluorides; Cl⁻: Chlorides; Cl₂: Residual chlorine; HCO₃⁻: Bicarbonates; STD: Total dissolved solids; MPL: maximum permissible limit, N.E: not specified.

The hardness in the water was classified by the World Health Organization [12], based on the concentrations of $CaCO_3$ (mg/L), such as: soft (0–60), moderately hard (61–120), hard (121–180) and very hard (>180). During the study conducted in the Lerma-Chapala Basin it was found that in 47.37% of the wells, located in the municipalities of Tenango del Valle, Toluca and El Oro, the water is moderately hard; 42.11% of the wells located in the municipalities of San Felipe del Progreso, Tenango del Valle, Toluca and El Oro, the water is hard and in 10.53% of the wells, located in the municipalities of San Felipe del Progreso and Tenango del Valley, the water is very hard. Furthermore, the highest value of alkalinity, well 11 (387.04 mg/L), exceeds the maximum limit allowed by the official Mexican standard (NOM-127-SSA1-2000); the smallest value was found in well 23 (87.08 mg/L), located in the municipality of El Oro.

In the Valley of Mexico Basin, the concentration of hardness showed that in 37.5% of the wells, located in the municipality of Zumpango de Ocampo, the water is hard; in 62.5% of the wells located in the municipalities of Cuautitlán Izcalli and Zumpango de Ocampo, it is very hard. According to the type of water found in this study, it was detected that with respect to soft and moderately hard water, it can be assumed that there is a minimum amount of dissolved salts, the hard and very hard waters contain a high volume of chemical elements such as Calcium and Magnesium [26].

In the Lerma-Chapala basin (Table 2), the $HCO_3^-$ concentration ranged between 106 and 472 mg/L, with a mean value of 198 mg/L. However, in the Valley of Mexico basin (Table 1) the highest value reached 781.47 mg/L (well 8), in the municipality of Zumpango de Ocampo and the lowest value was of 408.27 mg/L (well 9); that is, 40% and 74% higher, respectively, than those found in the Lerma-Chapala basin (Figure 3). The presence of bicarbonates in the water can be due to the presence of $CO_2$, at a $CO_3^{2-}$ dilution, or to the alteration of silicates [24,27,28]. The high content of bicarbonate ions, in both basins, can be associated to a great extent with a process of alteration of silicates, coming from the feldspar minerals, this alteration is carried out by the charge of the hydrogen ions towards the minerals releasing the silicates [29,30]. In addition to the silicate weathering, anthropogenic sources may also be responsible for the variation of groundwater chemistry in the study area. Likewise, the source of $Cl^-$ ions could have been derived from anthropogenic activities such as agricultural activities, this last because the highest total dissolved solids (TDS) value was found in the Valley of Mexico basin, in well 5 (1050.9 mg/L), these high values of TDS may be related to an ion exchange process and possible agricultural activity [31].

The spatial distribution of the $F^-$ ions is shown in Figure 3. The wells that show the highest concentration of these ions are located towards the center of the Valley of Mexico basin, far from the recharge zones and in a well close to the area of recharge of the Lerma-Chapala Basin, one of the activities carried out in the zones is agriculture, which could indicate that the source of fluorine in the water is associated with this activity. The evolution of the presence of the $F^-$ ions is observed towards the center of the basins and specifically in the Lerma-Chapala Basin in the direction of the surface water flow (Lerma River). According to the classification established by the WHO, the content of fluoride ions in the wells located in the Lerma-Chapala Basin, mainly the wells from San Juan Zitlaltepec, San José La Loma, Santiaguito, San Juan (municipality of Zumpango de Ocampo) and San Pedrito (Tenango del Valle) is considered medium to high [32]. In 1987, Alanis, in the study conducted in 108 municipalities of the State of Mexico, reported low values of fluorides (0 to 29 mg/L), medium (0.3 to 0.69 mg/L), optimal (0.7 to 0.99 mg/L) and high (1.0 to 1.5 mg/L); that is, they do not exceed the maximum limit allowed by standard NOM-127-SSA1-2000 (1.5 mg/L) (Table 2). Referring to the classification carried out by the WHO, in this study it was found that 78.95% of the wells have a low content of $F^-$ ions, 15.79% a medium content of $F^-$ ions and 5.26% a high content of $F^-$ ions, the highest value found in this basin was 1.31 mg/L [17,18].

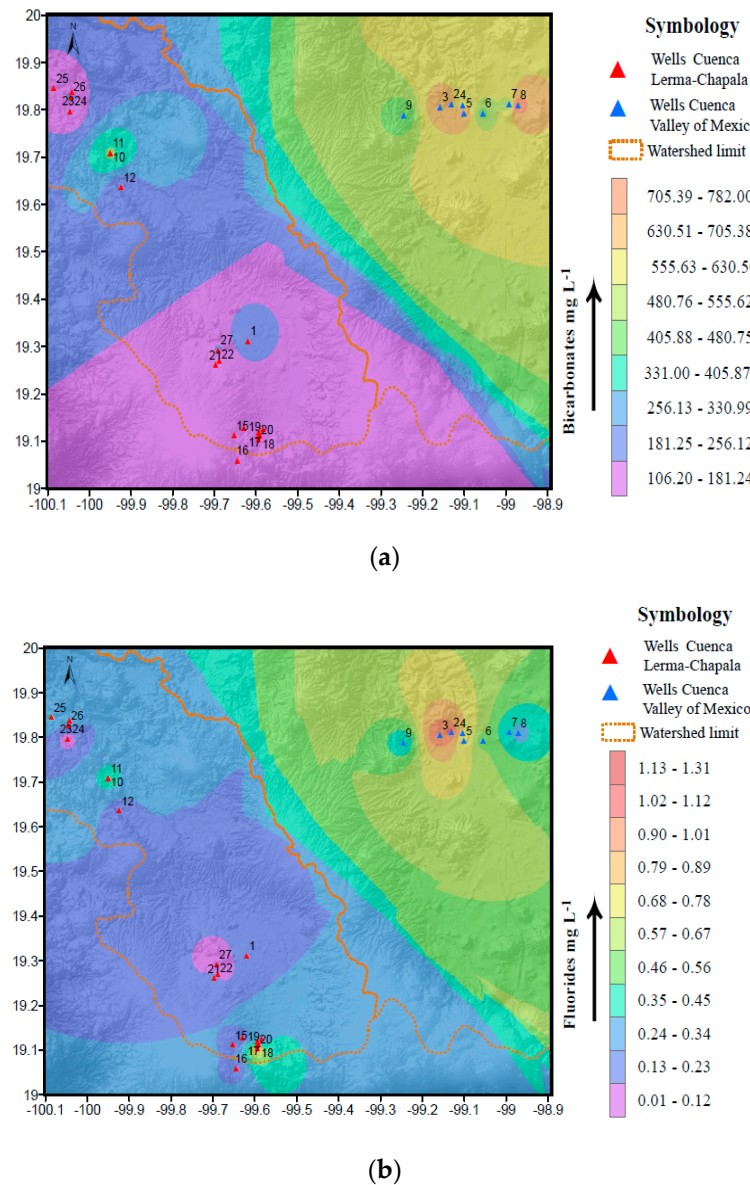

**Figure 3.** (**a**) Spatial distribution of Bicarbonate from 27 groundwater samples from the Lerma—Chapala and Valley of México basins; (**b**) Spatial distribution of Fluoride ions from 27 groundwater samples from the Lerma—Chapala and Valley of México basins.

In the Valley of Mexico basin, it can be observed that distribution of $F^-$ ions content is as follows: 12.5% of the wells of the is low range, 37.5% in the average range, 25% optimum range and in 25% a high content range (Table 1). It has been mentioned that when the filtration water is rich in calcium ions, water of the Ca-HCO$_3$ type, and comes into contact with a material rich in Na$^+$, the ion exchange process is observed in magnitude, releasing Na$^+$ ions and favoring the release of $F^-$, so that the water that reaches the aquifer is of the Na-HCO$_3$ type, and therefore, the concentration of $F^-$ ions tends to be greater than 1.0 mg/L. In contrast, when the water in the aquifer is of the Ca-HCO$_3$ type, the concentration of $F^-$ ions tends to be less than 1.0 mg/L [17,28,33–35].

According to this behavior and the data obtained from samplings (Tables 3 and 4) that the water in the aquifers of the Lerma-Chapala basin is of the Ca-HCO$_3$ type (Figures 4 and 5) [24,28,36]; however, for water from wells in the Valley of México basin, 75% could be of the Na-HCO$_3$ type and 25% of the Ca-HCO$_3$ type. Thus, it is possible to say that the hardness in the water of the Lerma-Chapala basin is a mostly due to calcium and in the Valley of México basin only 25% of the wells have calcium hardness.

The F⁻ ion presence can be from soil contamination as a result of the use of phosphate fertilizers and pesticides leached into the aquifer. As of this date, there is no documented or recorded health risk due to exposure to F⁻ in the study area.

**Table 3.** Concentrations of $Ca^{2+}$, $Mg^{2+}$, $Na^+$, $K^+$, $SO_4^{2-}$ and $NO_3^-$ ions of water samples from the wells with higher fluoride ion content, located in the Valley of Mexico basin.

| ID | $Ca^{2+}$ | $Mg^{2+}$ | $Na^+$ | $K^+$ | $SO_4^{2-}$ | $NO_3^-$ |
|---|---|---|---|---|---|---|
| | (mg/L) | (mg/L) | (mg/L) | (mg/L) | (mg/L) | (mg/L) |
| 2 | 14.81 | 18.71 | 31.00 | 6.05 | 71.59 | 6.3 |
| 3 | 19.04 | 11.77 | 19.40 | 3.74 | 43.63 | 7.8 |
| 4 | 19.04 | 96.85 | 161.53 | 31.47 | 67.92 | 9.2 |
| 5 | 10.58 | 48.76 | 81.19 | 15.79 | 108.66 | 4.2 |
| max | 19.04 | 96.85 | 161.53 | 31.47 | 108.66 | 9.2 |
| min | 10.58 | 11.77 | 31.00 | 6.05 | 43.63 | 4.2 |
| average | 15.87 ± 4.05 | 44.02 ± 38.71 | 73.28 ± 64.66 | 14.25 ± 12.61 | 72.95 ± 26.85 | 6.88 ± 2.14 |

**Table 4.** Concentrations of $Ca^{2+}$, $Mg^{2+}$, $Na^+$, $K^+$, $SO_4^{2-}$ and $NO_3^-$ ions of water samples from the wells with higher fluoride ion content, located in the Lerma—Chapala basin.

| ID | $Ca^{2+}$ | $Mg^{2+}$ | $Na^+$ | $K^+$ | $SO_4^{2-}$ | $NO_3^-$ |
|---|---|---|---|---|---|---|
| | (mg/L) | (mg/L) | (mg/L) | (mg/L) | (mg/L) | (mg/L) |
| 16 | 8 | 2.43 | 3.8 | 0.701 | 6.62 | 6.2 |
| 18 | 10 | 4.13 | 6.65 | 1.25 | 13.42 | 1.4 |
| 19 | 10 | 5.59 | 9.08 | 1.73 | 7.07 | 7.8 |
| 20 | 22 | 3.89 | 6.24 | 1.18 | 27.67 | 12 |
| 23 | 14 | 6.32 | 10.3 | 1.97 | 8.13 | 6.6 |
| 24 | 18 | 8.75 | 14.36 | 2.76 | 6.01 | 2.2 |
| 25 | 20 | 12.88 | 21.27 | 4.11 | 9.1 | 5.6 |
| 26 | 20 | 18.23 | 30.2 | 5.85 | 26.07 | 2.5 |
| max | 22 | 18.23 | 30.2 | 5.85 | 27.67 | 12 |
| min | 8 | 2.43 | 3.8 | 0.7 | 6.01 | 1.4 |
| average | 16.14 ± 5.76 | 7.24 ± 5.24 | 11.84 ± 8.75 | 2.27 ± 1.71 | 13.01 ± 8.86 | 5.54 ± 3.50 |

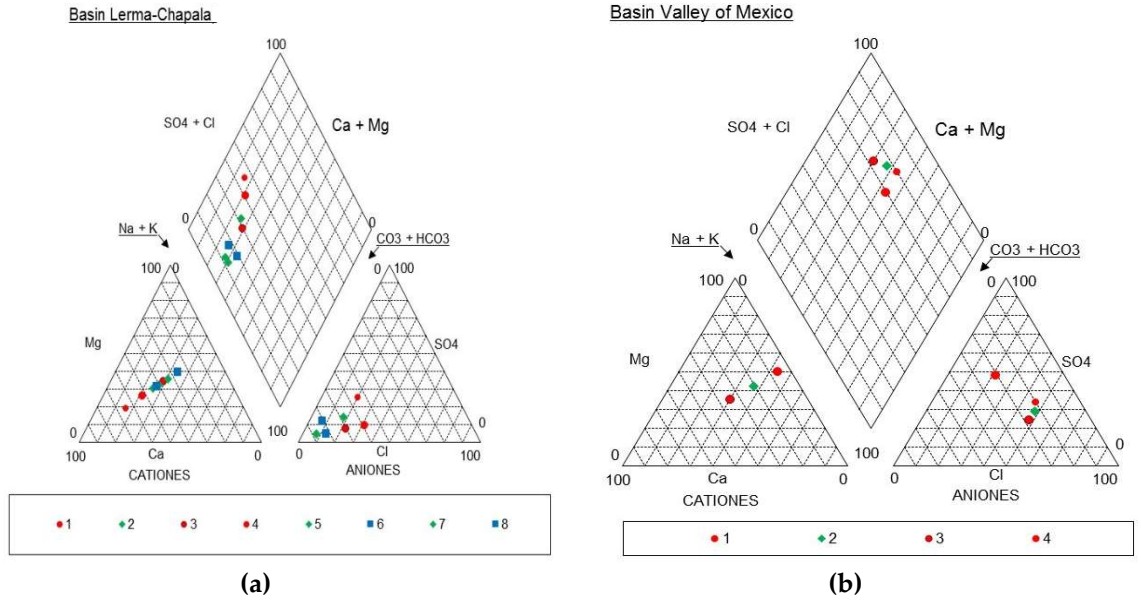

**Figure 4.** Piper diagram, water type classification of (**a**) the Lerma—Chapala basin and (**b**) the Valley of Mexico basin.

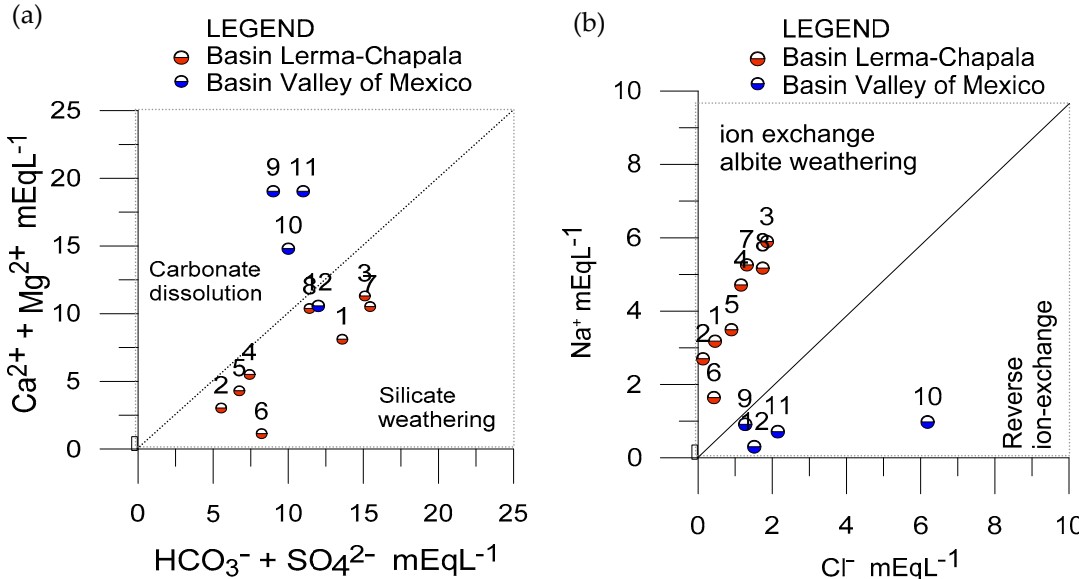

**Figure 5.** Relationship between concentration of (**a**) ($Ca^{2+} + Mg^{2+}$) and ($HCO_3^- + SO_4^{2-}$), (**b**) $Na^+$ and $Cl^-$.

Analyzing the concentrations of $F^-$ ions and hardness, found in the aquifers of this basin (Table 5), we observe an inverse relationship (R = −0.69) between both parameters; that is, in the wells where high hardness values are reached (200 to 460 mg/L) the concentration of $F^-$ ions is less than 1.0 mg/L; however, wells with hardness values between 120 and 200 mg/L, the concentration of $F^-$ ions is greater than 1.0 mg/L (R = −0.88). The correlation analysis of the above-mentioned parameters (Table 5) shows a significant and strong relationship between the fluoride ions and electrical conductivity (R = 0.5933), alkalinity (R = 0.6924), bicarbonates (R = 0.6924) and TDS (R = 0.5933) (Figure 6). While the relationship between fluoride ions and chloride ions and crude chlorine is shown inverse (R = −0.0208 and R = −0.3520, respectively). In addition, a relationship between electrical conductivity and calcium hardness is established, where it was observed that calcium ions control the electrical conductivity more than other ions.

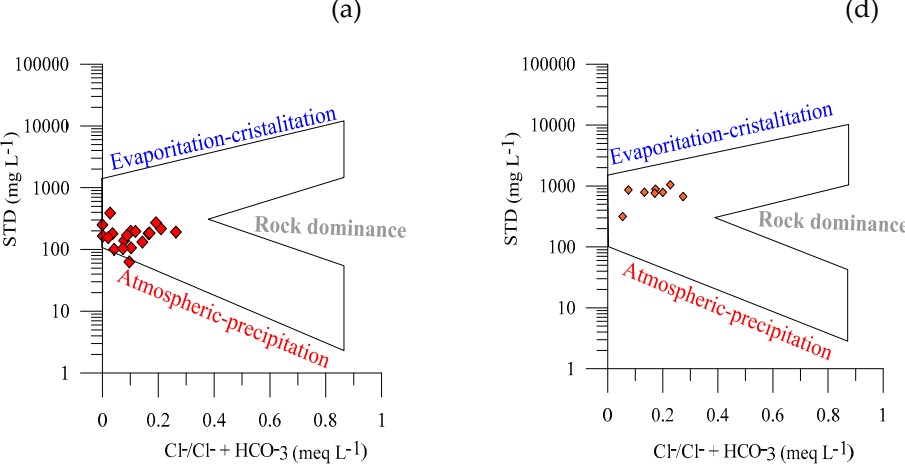

**Figure 6.** *Cont.*

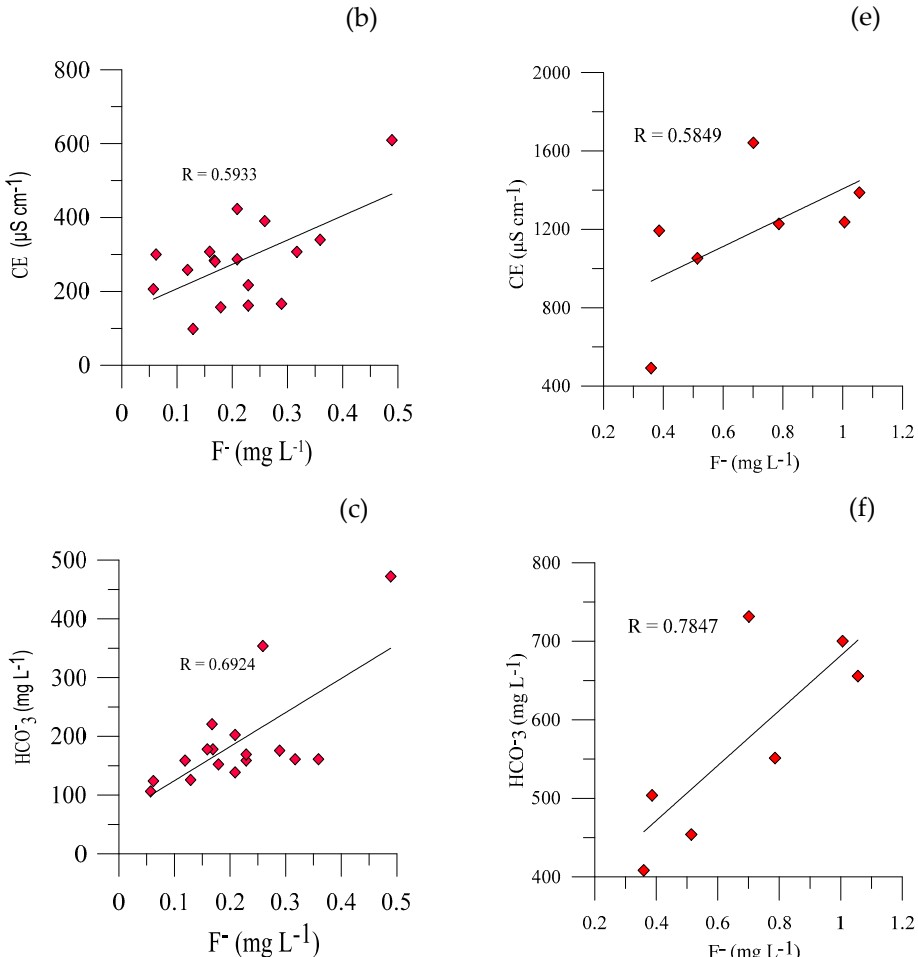

**Figure 6.** (**a**) Gibbs Diagram Cuenca Lerma-Chapala. TDS relationship $Cl^-/(Cl^- + HCO_3^-)$, identification of the process that control the chemistry of groundwater, (**b**) correlation EC and $F^-$, (**c**) correlation $HCO_3^-$ and $F^-$, (**d**) Gibbs Diagram Cuenca Valley of México, (**e**) EC correlation and $F^-$, (**f**) correlation $HCO_3^-$ and $F^-$.

**Table 5.** Correlation matrix of in situ parameters, fluoride and physicochemical ions of the water samples from the wells located in the Lerma-Chapala basin.

|  | $F^-$ | pH | EC | T | $Cl^-$ | Alkal | $Cl_2$ | Hard | $HCO_3^-$ | STD |
|---|---|---|---|---|---|---|---|---|---|---|
| $F^-$ | 1.0000 |  |  |  |  |  |  |  |  |  |
| pH | 0.0833 | 1.0000 |  |  |  |  |  |  |  |  |
| CE | 0.5933 | 0.2389 | 1.0000 |  |  |  |  |  |  |  |
| T | 0.1909 | 0.3172 | 0.4726 | 1.0000 |  |  |  |  |  |  |
| $Cl^-$ | −0.0208 | 0.1609 | 0.2091 | 0.1175 | 1.0000 |  |  |  |  |  |
| Alcal | 0.6924 | 0.1061 | 0.7881 | 0.2245 | −0.2051 | 1.0000 |  |  |  |  |
| $Cl_2$ | −0.3520 | −0.1225 | 0.4659 | 0.7574 | 0.4242 | −0.2310 | 1.0000 |  |  |  |
| Hard | 0.3367 | 0.3257 | 0.7778 | 0.2026 | 0.0947 | 0.7159 | 0.4018 | 1.0000 |  |  |
| $HCO_3^-$ | 0.6924 | 0.1061 | 0.7881 | 0.2245 | −0.2051 | 1.0000 | −0.2310 | 0.7159 | 1.0000 |  |
| STD | 0.5933 | 0.2389 | 1.0000 | 0.4726 | 0.2091 | 0.7881 | 0.4659 | 0.7779 | 0.7881 | 1.0000 |

Table 6 shows the correlation of fluoride ions with respect to temperature, pH, electrical conductivity, chlorides, alkalinity, free chlorine, hardness, bicarbonates and TDS. The correlations between fluoride ions, electrical conductivity, alkalinity, bicarbonates and TDS are positive; this allows us to observe that the content of fluoride ions present in groundwater is associated with high concentrations of bicarbonate ions and possibly the presence of sodium, due to a process of alteration of the silicates at the time of residence of the water. With regard to the presence of alkalinity, a strong

relationship with fluorides is observed which could be due to the presence of calcium salts and their control by ion exchange [37,38].

**Table 6.** Correlation matrix of in situ parameters, fluoride ions and physicochemical parameters of water samples from wells located in the Valley of Mexico basin.

| | F$^-$ | T | pH | CE | Cl$^-$ | Alkalinity | Hardness | HCO$_3$$^-$ | STD |
|---|---|---|---|---|---|---|---|---|---|
| F$^-$ | 1.0000 | | | | | | | | |
| T | 0.2247 | 1.0000 | | | | | | | |
| pH | −0.4236 | 0.0624 | 1.0000 | | | | | | |
| CE | 0.5493 | 0.4594 | −0.3998 | 1.0000 | | | | | |
| Cl$^-$ | 0.3525 | 0.8033 | −0.2157 | 0.8593 | 1.0000 | | | | |
| Alkalinity | 0.7884 | 0.2220 | −0.7585 | 0.8272 | 0.5695 | 1.0000 | | | |
| Hardness | −0.2153 | −0.4211 | −0.2490 | 0.4326 | 0.1138 | 0.2706 | 1.0000 | | |
| HCO$_3$$^-$ | 0.7884 | 0.2220 | −0.7585 | 0.8272 | 0.5699 | 1.0000 | 0.2706 | 1.0000 | |
| STD | 0.5493 | 0.4594 | −0.3998 | 1.0000 | 0.8593 | 0.8272 | 0.4326 | 0.8272 | 1.0000 |

In addition to the parameters analyzed in situ, the concentration of residual chlorine was determined before the disinfection process. In 52.6% of the wells, chlorine was observed, with values ranging between 0.3 and 1.3 mg/L; this parameter was called crude chlorine (Table 2). Crude chlorine may be present in the aquifer due to a reduction process that involves the presence of organic matter. In 2019, in the study conducted by Fonseca, in some wells of the Toluca Valley; evidence was obtained of the presence of crude chlorine, mainly in those wells with anthropogenic organic matter content [28].

In the Alanis study (1987), the reported concentrations of fluoride ions helped to observe if there had been any significant variation in some sampling sites over time [32]. With this study it can be observed that there is no significant difference with respect to time, in some sampling sites the concentration shows a decrease in fluoride ions; this behavior is important because a low content would mean a low contribution of fluorides to humans when consuming this water. CONAGUA facilitated a report made in 1997 by the Institute of Health of the State of Mexico (ISEM), where it proposes to the population of the municipalities of San Felipe del Progreso, Sultepec, Tenango del Valle, Zumpango, Cuautitlan Izcalli, Ecatepec, Texcoco and Tlalnepantla not to consume fluoridated salt in its daily diet, because these areas are considered of high risk for fluorosis; this may be due to the fact that the content of fluoride ions in the groundwater depends on the depth at which the water is extracted, geological characteristics of the place and the time of contact of the water with the ground.

Using a Gibbs diagram, the processes that could be occurring in the aquifer were analyzed; according to the diagram, relationship of TDS and Cl$^-$/(Cl$^-$ + HCO$_3$$^-$) (Figure 6a). It is observed that the water from the aquifer of the Lerma-Chapala basin comes from meteoric precipitation (rain and its rapid infiltration), with lateral dispersion associated with water-rock interaction; this suggests that the control mechanism of the groundwater chemistry is due to the high rainfall that reaches 1300 mm/year [23] and the weathering of volcanic rocks, mainly silicates. Samples show HCO$_3$$^-$ > Cl$^-$; however, the larger content of Cl$^-$ was observed in well 26, this could be associated with infiltration of wastewater, associated with the local recharge and the greater interaction is presented by the sample 22. According to the data found, in the Lerma-Chapala basin, it is possible to consider that the local infiltration water is of the CaHCO$_3$ type, little evolved. Water from the aquifer of the Valley of Mexico is located in the area of evaporation with lateral dispersion and little evolution. This behavior is observed in the farming areas that receive subsurface runoff by agricultural return waters, raising the sodium and TDS content.

## 4. Conclusions

The physicochemical parameters in situ showed great spatial variation, except T and pH. The low values of temperature and pH were identified near the recharge zones and increase in the direction of

the groundwater flow. The water analyzed in the Lerma Chapala basin is neutral to slightly alkaline and in the Valley of Mexico basin it is mostly alkaline.

The high content of bicarbonate ions, in both basins, can be associated to a process of alteration of silicates, coming from the feldspar minerals. The source of $Cl^-$ ions could have been derived from anthropogenic activities, because the highest TDS value was found in the Valley of Mexico basin. High values of TDS can also be related to an ion exchange process and possible agricultural activity.

The highest EC value was found in the shallowest well, which suggests that the water is more mineralized at this point and that it may be receiving infiltrated surface water. In the Lerma-Chapala basin, the EC increases towards the central part parallel to the flow of groundwater.

The high content of $F^-$ ions in the Lerma-Chapala basin reached 1.31 mg/L. In the Valley of México basin, the highest value was 1056 mg/L (25% of the samples). Wells with high hardness values (200 to 460 mg/L) have $F^-$ ion concentration less than 1.0 mg/L; however, for wells with hardness values between 120 and 200 mg/L, the concentration of $F^-$ ions is greater than 1.0 mg/L.

The aquifer water from the Lerma-Chapala basin is of the Ca-HCO$_3$ type and the water from the wells of the Valley of México basin, of the Na-HCO$_3$ and Ca-HCO$_3$ types. Therefore, the hardness in the water of the Lerma-Chapala basin is mostly calcium, in the Valley of Mexico basin only 25% of the wells have calcium hardness. The source of ions $F^-$ can be soil contamination as a result of the use of phosphate fertilizers and pesticides that leach into the aquifer. Up to this date, there is no record of the health risk due to exposure to $F^-$.

**Author Contributions:** Investigation, J.L.G.-R., B.G.-G. and R.M.G.F.-M.d.O.; writing—original draft preparation, R.M.F.-R. and C.J.V.-A.; writing—review and editing, R.M.F.-R.; funding acquisition, J.L.G.-R.

**Funding:** This research received external funding from the Program for Professional Development Teacher, for the Superior Type (PRODEP) DSA/103.5/16/10542.

**Acknowledgments:** The authors thankfully acknowledge the support of the Technological Institute of Toluca, the corresponding water and sanitation agencies of Cuautitlán Izcalli, El Oro, San Felipe del Progreso, Tenango del Valle, Toluca and Zumpango de Ocampo for the permissions granted for the development of this project. Ricardo Contreras Martínez of the National Institute of Statistics and Geography, Subdirectorate of Geography, Department of Basic Information and Consejo Nacional de Ciencia y Tecnología (CONACYT) for the scholarship given to C.J.V.-A.

**Conflicts of Interest:** The authors declare that they no conflicts of interest.

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
