# Peer review of "Presence and Distribution of Fluoride Ions in Groundwater for Human in a Semiconfined Volcanic Aquifer"

_resources, doi:10.3390/resources8020116_

Round 1

Reviewer 1 Report

The purpose of this study was to determine the presence and distribution of fluoride ions in the groundwater for human use and consumption in six municipalities of the State of Mexico, four located in the aquifers within the limits of the Lerma-Chapala Basin and two in the basin of the Valley of Mexico, and to establish the health risk from its consumption.

The correlation analysis shows a significant and strong relationship between the fluoride ions vs. alkalinity (R = 0.6924) and TDS (R = 0.5933), which is of interest.

The article is relevant, but requires in-depth processing and re-submission.

A more extensive discussion of the relationship between fluorine concentration, the chemical composition of groundwater, the lithologic composition of aquifers and the depth of wells is needed.

Minor comments:

2. Materials and Methods

2.1. Study area

You write:

The water from the Lerma-Chapala basin supports human as well as agricultural and industrial needs. Within this basin, specifically in the jurisdiction of the State of Mexico, are the aquifers 1501 (Valley of Toluca), 1502 (Ixtlahuaca-Atlacomulco), 1504 (Tenancingo) and 1505 (Villa Victoria-Valley of Bravo), these aquifers feed the underground zone of the municipalities of El Oro, San Felipe del Progreso, Tenango del Valle and Toluca (Figure 1). 12.7 °C.

As for the aquifer flow system of the Toluca Valley, it is formed by three main directions, the first originates from the Sierra de las Cruces towards the foothills west-circulation, the second comes from the Nevado de Toluca North direction with exit at Ixtlahuaca, and the third originates from the infiltrations of the Nevado east-northeast direction [20].

The aquifer 1508 is found in this basin (Cuautitlán-Pachuca), which feeds the underground zone of the municipalities of Cuautitlán Izcalli and Zumpango de Ocampo (Figure 1).

What do the numbers 1502, 1504, 1505, 1508 mean?

What does the temperature "12.7 °C" mean?

Fig. 1.

Need to show:

- names of rivers,

- boundaries of “body water” (H2O),

- boundaries of aquifers 1502, 1504, 1505, 1508- three main flow directions

Instead of Simbology, there should be a Legend.

It is advisable to present three hydrogeological cross-sections in the following directions: samples 25-12, 27-16, 9-8 with lithology and sampling intervals

2.2. Sampling and characterization of groundwater

It is necessary to provide a table with the characteristics of 27 samples: Well Name, Bore hole depth (m), Aquifer (sand, andesites, basalts, etc) – see example: https://doi.org/10.1016/j.ejrh.2018.09.003

3. Results and discussions

You write:

alkalinity (as CaCO3)

Table 1.

Need to clarify the meaning: «Pozo, LMP, S/C, N.E, Promedio»

Instead of STD, there should be a TDS.

You write:

a high volume of minerals such as Calcium. and Magnesium

These are chemical elements, not minerals.

Author Response

Dear Reviewer,

I send you the corrected manuscript of the article entitled "
Presence and distribution of fluoride ions in groundwater for human in a semiconfined volcanic aquifer". The changes were made according your comments.

1)      The correlation analysis shows a significant and strong relationship between the fluoride ions vs. alkalinity (R = 0.6924) and TDS (R = 0.5933), which is of interest.

The correlation analysis shows the significant relationship between F- vs CE (R = 0.5933), F- vs alkalinity (R = 0.6924), F- vs HCO3- (R = 0.6924) and F- vs TDS (R = 0.5933), this could confirm that, the content of fluoride ions present in groundwater is associated with high concentrations of bicarbonate ions and possibly the presence of sodium, due to a process of alteration of the silicates.

2)      2. Materials and Methods

            2.1. Study area

            What do the numbers 1502, 1504, 1505, 1508 mean?

The numbers 1502, 1504, 1505 and 1508 are codes assigned by the national water commission in Mexico, these codes are only for their classification therefore they were removed from the document.

3)      Fig. 1.

            Need to show:

            - names of rivers,

            - boundaries of “body water” (H2O),

            - boundaries of aquifers 1502, 1504, 1505, 1508- three main flow directions

              Instead of Simbology, there should be a Legend.

Figure 1 is being corrected, for the time of delivery of the revisions the co-author in charge did not conclude on time, I ask you please to give me the opportunity to change it in the next revision.

4)      It is advisable to present three hydrogeological cross-sections in the following directions: samples 25-12, 27-16, 9-8 with lithology and sampling intervals

The hydrogeological analysis was not carried out in the work, we do not have this information

5)      It is necessary to provide a table with the characteristics of 27 samples: Well Name, Bore hole depth (m), Aquifer (sand, andesites, basalts, etc)

In tables 1 and 2 the depth data were appended, there is no information on the name of the wells and the statigram column.

6)      3. Results and discussions  You write:  alkalinity (as CaCO3)

The change was made: alkalinity (CaCO3)

7)      Table 1.  Need to clarify the meaning: «Pozo, LMP, S/C, N.E, Promedio»

Instead of STD, there should be a TDS.

The change was made

8)      This paragraph should go at the beginning where are you describing the study area. You write: a high volume of minerals such as Calcium and Magnesium. These are chemical elements, not minerals.

The change was made

Reviewer 2 Report

Dear authors,

The article Presence and distribution of fluoride ions in groundwater for human in a semiconfined volcanic aquifer deals with very interesting subject but have some weakness in the discussion that is need to be improved. My comments and suggestions are given in the attachment.

Author Response

Dear Reviewer,

I send you the corrected manuscript of the article entitled "
Presence and distribution of fluoride ions in groundwater for human in a semiconfined volcanic aquifer". The changes were made according your comments.

1)    Abstract: Dental and emaciated fluorosis is derived from the chronic intake of fluoride ions (F-) by consumption of food, tooth products and drinking water from underground aquifers, change by groundwater

The change was made

2)    rewrite the sentence The water of the Toluca Valley aquifer is located in the meteoric precipitation field, with lateral dispersion associated with the water-rock interaction.

According to Gibbs diagram, the water of the Toluca Valley aquifer is recharged by local meteoric precipitation and consequent the water-rock interaction occurs.

3)    In Mexico, the main source of water supply for human consumption is from underground origin; change by groundwater

The change was made

4)    Typical results, In the municipality of Cuautitlán Izcalli, change by in

The change was made

5)    The region of the Lerma-Chapala basin is located in the center west of the Mexican republic with an extension is 53 591 km2, change by Km2

The change was made

6)    Rewritten the paragraph “The Toluca Valley aquifer is made up of a porous geological medium and a fractured medium; the porous medium is formed by unconsolidated clastic deposits, that fill the basin of the Toluca Valley; represented by tuffs, sands, alluvial deposits (Qal), lacustrine deposits (Qla), the Tarango formation, flows from the Nevado de Toluca and recent volcanic cones; the fractured medium is formed of andesites (Tpv), and ancient basalts. With these porosity geological properties, the aquifer is integrated as a free upper aquifer and the fractured medium as a semiconfined lower aquifer. Even when both layers are divided by impermeable materials, they are considered part of the same aquifer”.

According the porosity the Toluca Valley aquifer is divided in two types: porous and fractured. The porous aquifer is build up from unconsolidated clastic deposits, that fill the basin of the Toluca Valley, represented by tuffs, sands, alluvial deposits (Qal), lacustrine deposits (Qla) of the Tarango formation. The aquifer with fractured porosity is build up from andesites (Tpv), and ancient basalts. With these porosity geological properties, the aquifer is integrated as a free upper aquifer and the aquifer  with fractured medium as a semiconfined lower aquifer

7)    This paragraph should go at the beginning where are you describing the study area.

The Valley of Mexico Basin is located in the center-south of the Transverse Neovolcanic Axis, has an average altitude of 2240 m above sea level and an approximate area of 9600 km2. It is limited to the north by the Sierra de Pachuca and the Sierra de Tezontlalpan, to the south by the Sierra de Chichinautzin, to the east by the Sierra Nevada, the Sierra Calpulalpan and the Sierra de Tepozán, and to the west by the Sierra of Monte Bajo and the Sierra de las Cruces. The aquifer 1508 is found in this basin (Cuautitlán-Pachuca), which feeds the underground zone of the municipalities of Cuautitlán Izcalli and Zumpango de Ocampo (Figure 1).

The change was made

8)    2.2. Sampling and characterization of groundwater

The change was made

9)    Change to English: pozo, maximo, minimo and promedio in the tables

The change was made

10) There is general problem with discussion and at the with conclusions. measurements of Na, K, Ca, Mg and SO42 is missing and you are describing the water types and etc based upon published articles. and you are afterwards making some conclusion upon data that you did not measure. or you did?looking at your references i have feeling  that you did measure but you did not want to publish here because you already published somewhere else. but, to proof some statement in this article you have to have this data. for example Na -Cl vs Ca+Mg - Cl+SO42 and CaMg vs HCO3 can proof silicate weathering, influence of anthropogenic activities etc. so, please ad this and make with this data improvement of the article's discussion and conlusions.

The data corresponding to the type of water, the piper diagrams and the suggested graphs were attached

Round 2

Reviewer 1 Report

The purpose of this study was to determine the presence and distribution of fluoride ions in the groundwater for human use and consumption in six municipalities of the State of Mexico, four located in the aquifers within the limits of the Lerma-Chapala Basin and two in the basin of the Valley of Mexico, and to establish the health risk from its consumption.

The correlation analysis shows a significant and strong relationship between the fluoride ions vs. alkalinity (R = 0.6924) and TDS (R = 0.5933), which is of interest.

The article is relevant, but You write: Figure 1 is being corrected, for the time of delivery of the revisions the co-author in charge did not conclude on time, I ask you please to give me the opportunity to change it in the next revision.

Minor comments:

Fig. 1.

Need to show:

- names of rivers,

- boundaries of “body water” (H2O),

- boundaries of aquifers 1502, 1504, 1505, 1508- three main flow directions

Author Response

Dear Reviewer,

I send you the corrected manuscript of the article entitled "
Presence and distribution of fluoride ions in groundwater for human in a semiconfined volcanic aquifer". The changes were made according your comments.

1)      Fig. 1.

            Need to show:

            - names of rivers,

            - boundaries of “body water” (H2O),

            - boundaries of aquifers 1502, 1504, 1505, 1508- three main flow directions

              Instead of Simbology, there should be a Legend.

The change was made
